# Postmortem Quantitative Analysis of Prion Seeding Activity in the Digestive System

**DOI:** 10.3390/molecules24244601

**Published:** 2019-12-16

**Authors:** Katsuya Satoh, Takayuki Fuse, Toshiaki Nonaka, Trong Dong, Masaki Takao, Takehiro Nakagaki, Daisuke Ishibashi, Yuzuru Taguchi, Ban Mihara, Yasushi Iwasaki, Mari Yoshida, Noriyuki Nishida

**Affiliations:** 1Department of Locomotive Rehabilitation Science, Nagasaki University Graduate School of Biomedical Sciences, Nagasaki 852-8501, Japan; trangneuro@gmail.com; 2Department of Molecular Microbiology and Immunology, Nagasaki University Graduate School of Biomedical Sciences, Nagasaki 852-8501, Japan; takafuse-ngs@umin.ac.jp (T.F.); t.neuron.bbbapoe2012@gmail.com (T.N.); rb79ball2@hotmail.com (T.N.); dishibashi-ngs@umin.ac.jp (D.I.); ytaguchi@nagasaki-u.ac.jp (Y.T.); noriyukinishida@icloud.com (N.N.); 3Department of Neurology, Institute of Brain and Blood Vessels, Mihara Memorial Hospital, Isesaki 372-0006, Japan; msktakaobrb@gmail.com (M.T.); ban.mihara@mihara-ibbv.jp (B.M.); 4Department of Neurology International Medical Center, Saitama Medical University, Saitama 350-1298, Japan; 5Department of Neuropathology, Institute for Medical Science of Aging, Aichi Medical University, Aichi 480-1195, Japan; iwasaki@sc4.so-net.ne.jp (Y.I.); myoshida@aichi-med-u.ac.jp (M.Y.)

**Keywords:** prion, rt-quic

## Abstract

Human prion diseases are neurodegenerative disorders caused by prion protein. Although infectivity was historically detected only in the central nervous system and lymphoreticular tissues of patients with sporadic Creutzfeldt-Jakob disease, recent reports suggest that the seeding activity of Creutzfeldt-Jakob disease prions accumulates in various non-neuronal organs including the liver, kidney, and skin. Therefore, we reanalyzed autopsy samples collected from patients with sporadic and genetic human prion diseases and found that seeding activity exists in almost all digestive organs. Unexpectedly, activity in the esophagus reached a level of prion seeding activity close to that in the central nervous system in some CJD patients, indicating that the safety of endoscopic examinations should be reconsidered.

## 1. Introduction

Prion diseases are characterized by the accumulation of abnormal prion protein (protease-resistant prion protein (PrP-res)) in the central nervous system (CNS), and this accumulation is thought to be the sole cause of these diseases. Human prion diseases (HPDs) present in various forms, including Creutzfeldt-Jakob disease (CJD), Gerstmann-Sträussler-Scheinker syndrome (GSS), fatal familial insomnia, and kuru. CJD may be sporadic, genetic, or acquired. Iatrogenic CJD has been caused by reusing contaminated neurosurgical instruments or using biomaterials such as growth hormone and dura mater grafts [1,2,3]. Epidemiological investigations have concluded that sporadic CJD (sCJD) is rarely caused by the accidental transmission of prions from biological materials taken from the CNS [4]. In a recent study, however, PrP-res was detected in the spleen of a patient with sCJD; the level of abnormal PrP [scrapie isoform of the prion protein (PrP^Sc^)] in the spleen of this patient was approximately 10^−4^ higher than that in the brain tissue [5]. Because the conventional detection methods for PrP-res and infectivity of human prions using animal models are not sensitive enough, the exact distribution of human prions in patients with sCJD remains unknown.

Prion seeding activity can be determined by novel techniques for in vitro amplification of prions, such as the real-time quaking-induced conversion (RT-QuIC) assay. This is a highly sensitive and specific method to detect a very small amount of prion seeding activity [6]; the detection limit of our RT-QuIC assay is around 0.12 fg of PrP-res [7]. Recent studies have shown that seeding activity in vitro, determined by endpoint RT-QuIC, parallels the infectivity of prion-containing animal specimens. This allowed us to directly determine the 50% seeding dose (SD_50_) of human prions with good correlation to the infectivity (50% lethal dose) by endpoint titration of RT-QuIC. We previously reported that seeding activity is present in peripheral organs, including the liver, lung, and kidney, of patients with sCJD.

Some research groups have performed reanalyses using quantitative RT-QuIC for highly sensitive detection and quantification of PrP^Sc^ in prion-infected tissues [8]. PrP^Sc^ were detected at various time points during the incubation period in peripheral organs (spleen, heart, muscle, liver, and kidney) in two experimental scrapie strains (RML and ME7) in mice [9]. In addition, mock-inoculated animals show detectable skin/brain PrP^Sc^ only after long cohabitation periods with scrapie-infected animals, unlike with 263K-inoculated animals [9].

RT-QuIC testing of olfactory epithelium samples obtained from nasal brushings was recently shown to be accurate in diagnosing CJD and indicated substantial prion seeding activity lining the nasal vault [10]. The specimens necessary to perform RT-QuIC testing for diagnosis of CJD in human patients were nasal tissues and peripheral nerves [11].

Patients with the terminal stage of CJD experience difficultly eating meals on their own, and some Japanese patients with CJD thus undergo percutaneous endoscopic gastrostomy. The whole-body distribution of prions must be elucidated to reduce the risk of accidental prion infection in such patients. We herein report the activity of prions in the digestive organs of patients with sCJD and other peripheral tissues from patients with genetic CJD.

## 2. Results

### 2.1. Analysis of Extraneural Tissues from Patients with sCJD and Healthy Subjects Using Endpoint RT-QuIC Assay

The homogenized samples of eight different organs showed positive reactions in RT-QuIC with the exception of the duodenum and transverse colon from Patient 4 (sCJD), sigmoid colon from Patient 2 (sCJD), and esophagus from Patient 3 (sCJD) (Table 1). We tested homogenized samples three times with eight replications for accurate determination of the SD_50_. This approach was used because more than eight replications were needed when we evaluated the accuracy of our endpoint assay in RT-QuIC using a 10^−4^ diluted duodenal sample. This fits well with the theoretical number induced by Poisson’s distribution. Prion seeding activity in the esophagus was 10^8.38 ± 0.16^ in Patient 1 and 10^7.98 ± 0.16^ in Patient 2, whereas the RT-QuIC assay was negative in the sigmoid colon of Patient 1, jejunum and appendix of Patient 3, and stomach of Patient 4. The range of SD_50_ was 10 [5,6,7] for all digestive tissues from the four patients (Figure 1 and Figure 2).

### 2.2. Analysis of Extraneural Tissues from a Patient with GSS and a Patient with Genetic HPD Using Endpoint RT-QuIC Assay

The homogenized samples of eight different organs showed positive reactions in RT-QuIC with the exception of the sigmoid colon from a patient with GSS (Table 2). Prion seeding activity was 10 [6,7] in the esophagus, stomach, duodenum, and jejunum; 10^8.33 ± 0.67^ in the appendix; and 10^8.46 ± 0.44^ in the transverse colon.

The homogenized samples of 10 different organs showed positive reactions in RT-QuIC with the exception of the gastroesophageal junction from Patient 6 (genetic CJD). Prion seeding activity was 10 [7,8] in the esophagus, stomach, duodenum, and jejunum; 10^8.84 ± 0.23^ in the terminal ileum; and 10^8.21 ± 0.34^ in the stomach (Table 3).

### 2.3. Immunohistochemical Staining of the Gut

Immunohistochemistry for PrP-res revealed no abnormal PrP in any of the gut tissues.

## 3. Discussion

We evaluated prion seeding activity and determined the SD_50_ in digestive tissues of patients with HPD using the endpoint RT-QuIC assay. We previously determined that prion activity in peripheral solid organs of patients with sCJD reaches an SD_50_ of around 10^6^/g [12]. Unexpectedly, in the present study, the SD_50_ of digestive tissues reached > 10^6–8^/g. Except in Patient 3, the esophagus and appendix showed the highest titer. In patients with variant CJD, bovine spongiform encephalitis prion infection can be caused by oral intake of contaminated foods, and it is well documented that bovine spongiform encephalitis prions accumulate in the appendix and Peyer’s patches of the small intestine. However, the etiology of sCJD differs from that of variant CJD, and there is no evidence for oral infection of sCJD. Recent research has shown that RT-QuIC testing of olfactory epithelium samples obtained from nasal brushings is accurate in diagnosing CJD. Therefore, in patients with sCJD, secreted prions or nasal tissue might be repeatedly swallowed with resultant reabsorption of the prions in the gut. Diagnosis of sCJD might thus be possible by RT-QuIC testing of the upper digestive tract obtained from biopsy (GIS) from the esophagus, stomach, and duodenum.

Immunostaining revealed no abnormal PrP in any tissues. The SD_50_ in the transverse colon and cecum of the patient with GSS was 10^8.46^ and 10^8.33^, respectively, and that in the gallbladder and stomach of the patient with genetic HPD was 10^8.84^ and 10^8.21^, respectively. We assume that these levels of infectivity are lower than those in the CNS based on experimental infections in animals (data not shown). However, the fact that prion activity can be detected in some digestive organs should not be neglected because HPD can develop even after an incubation period of 30 to 40 years [13]. Furthermore, these findings indicate that we must be careful with surgical instruments when performing surgery involving the digestive organs (e.g., surgical treatment of colon cancer) in patients with genetic HPD.

Finally, Japanese doctors perform a gastrostomy for some patients with CJD. When clinicians create the gastrostomy, they can pass deep into the stomach and abdominal walls. Therefore, clinicians treat the gastrostomy carefully at the time of its creation [14].

## 4. Materials and Methods

### 4.1. Patients

Four CJD patients (three males; one female) who had been histopathologically diagnosed with classical-type sCJD (MM1 type) after autopsy were involved in this study. In addition, after autopsy, one patient was histopathologically diagnosed with GSS (P102L) and one patient was histopathologically diagnosed with genetic CJD (E200K) (Table 4). Organs were carefully harvested separately to avoid contamination of brain and lymphatic tissue. Tissue specimens were immediately stored at −80 °C until use. Western blot analysis of the proteinase K-resistant PrP-res fragment and genotyping at codon 129 of the *PRNP* gene were conducted as described elsewhere [6] by the reference laboratory of the Japan CJD Surveillance Unit. Non-CJD tissues were purchased from ProteoGenex, Inc. (Culver City, CA, USA). The study protocol was approved by the Ethics Committee of Nagasaki University Hospital (ID: 100428423), and ethical approval for the use of specimens was granted by the Japan CJD Surveillance Unit. The study was registered with the University Hospital Medical Information Network (ID: UMIN000003301& UMIN000038398).

### 4.2. Tissue Homogenate Preparation

Brain, esophagus, stomach, duodenum, jejunum, appendix, transverse colon, and sigmoid colon specimens were subjected to RT-QuIC to evaluate the SD_50_. To prevent contamination of brain to other samples, we used single-use disposable tubes and beads, and all procedures on the bench top were performed on different days. Tissue samples were homogenized at 10% (*w*/*v*) in ice-cold phosphate-buffered saline supplemented with a protease inhibitor mixture (Roche, Mannheim, Germany) using a multi-beads shocker (Yasui Kikai, Osaka, Japan). The samples were clarified by centrifugation at 6000 rpm for 2 min and stored at −80 °C. Histopathological analysis with hematoxylin and eosin staining and immunohistochemistry for PrP-res were performed as previously described [15].

### 4.3. Endpoint RT-QuIC

RT-QuIC was performed as previously described [16]. Each serially diluted sample was tested with eight replications, and PrP amyloid formation was monitored for 48 h. The assay was repeated at least three times, and the SD_50_ value was then calculated by the Spearman-Kärber method [17].

## 5. Conclusions

Activity in the esophagus reached a level of prion seeding activity close to that in the central nervous system in some CJD patients, indicating that the safety of endoscopic examinations should be reconsidered.

## Figures and Tables

**Figure 1 molecules-24-04601-f001:**
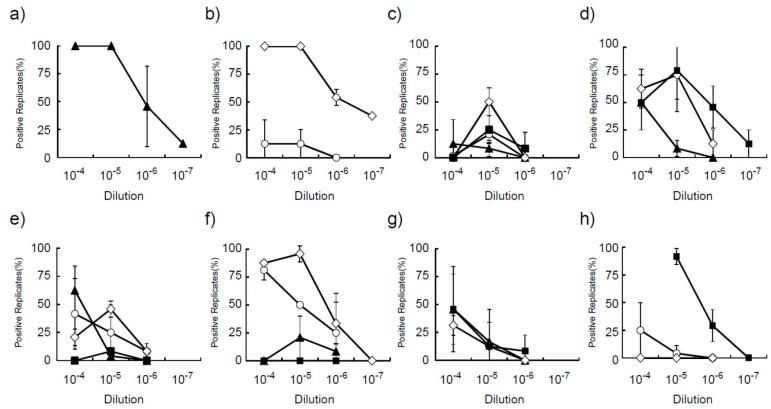
Ratio of positive replicates in diluted digestive organs of patients with sporadic Creutzfeldt-Jakob disease. The tissues were analyzed in the positive wells (*n* = 8) in the indicating dilution by real-time quaking-induced conversion. (**a**) Esophagus, (**b**) gastroesophageal junction, (**c**) stomach, (**d**) duodenum, (**e**) jejunum, (**f**) terminal ileum, (**g**) transverse colon, and (**h**) sigmoid colon from Patient 1 (open circle), Patient 2 (open diamond), Patient 3 (closed triangle), and Patient 4 (closed square) of three independent experiments.

**Figure 2 molecules-24-04601-f002:**
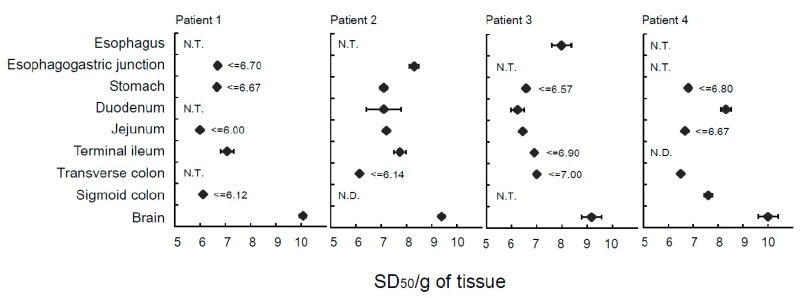
Seeding activity in digestive organs of patients with sporadic Creutzfeldt-Jakob disease. The SD_50_/g of digestive organ tissues was measured in four patients by real-time quaking-induced conversion and all prion seeding activities for digestive organs (SD_50_) were defined as log SD_50_/g of tissue. Data are presented as the mean ± standard deviation of three independent experiments as described in Table 1. The symbol “≤” means that the seeding activity was less than or equal to the indicated SD_50_. N.T., not tested; N.D., not detected; SD_50_, 50% seeding dose.

**Table 1 molecules-24-04601-t001:** Prion seeding activity of digestive organs in four sporadic Creutzfeldt-Jakob disease patients.

Tissue	Patient 1		Patient 2		Patient 3		Patient 4	
	Mean ^#3^	±S.D.	Mean	±S.D.	Mean	±S.D.	Mean	±S.D.
Esophagus	≤6.70		8.38	±0.16	7.98	±0.39	N.E.	
Stomach	≤6.50		7.10	±0.14	≤6.57		≤6.80	
Duodenum	N.E		7.10	±0.70	6.24	±0.25	8.31	±0.20
Jejunum	≤6.67		7.20	±0.12	6.44	±0.10	≤6.67	
Terminal ileum	7.07	±0.26	7.74	±0.25	≤6.90		N.D.	
Transverse colon	N.E.		≤6.14		≤7.00		6.50	±0.11
Sigmoid colon	≤6.12		N.D.		N.E.		7.60	±0.17
Brain ^#3^	10.08	±0.12	9.42	±0.12	9.17	±0.42	10.00	±0.35

All prion seeding activities for digestive organs (SD_50_) were defined as log SD_50_/g of tissue; N.D., not detected; ^#3^ SD_50_ of brain samples are provided in the report by Takatsuki et al. [7] An empty column indicates that no test was conducted. S.D., standard deviation; N.E. not examined.

**Table 2 molecules-24-04601-t002:** Prion seeding activity of digestive organs in one patient with Gerstmann-Sträussler-Scheinker syndrome.

Tissue	Patient 5	
	Mean	±S.D.
Esophagus	6.93	±0.32
Stomach	7.34	±0.20
Duodenum	≤6.57	
Jejunum	7.22	±0.30
Appendix	8.33	±0.67
Transverse colon	8.46	±0.44
Sigmoid colon	N.E.	
Brain	10.18	±0.34

All prion seeding activities for digestive organs (SD_50_) were defined as log SD_50_/g of tissue. S.D., standard deviation; N.E.

**Table 3 molecules-24-04601-t003:** Prion seeding activity of digestive organs in one patient with genetic Creutzfeldt-Jakob disease (E200K).

Tissue	Patient 6	
	Mean	±S.D.
Esophagus	8.5	±0.00
Stomach	8.21	±0.34
Duodenum	6.81	±0.31
Jejunum	≤6.33	±0.00
Ileum	7.17	±0.94
Appendix	7.23	±0.20
Cecum	≤6.64	
Transverse colon	≤5.80	
Sigmoid colon	7.47	±0.50
Brain	10.3	±0.35

All prion seeding activities for digestive organs (SD_50_) were defined as log SD_50_/g of tissue. S.D., standard deviation.

**Table 4 molecules-24-04601-t004:** Summary of patients with prion disease.

	Patient 1	Patient 2	Patient 3	Patient 4	Patient 5	Patient 6
	Sporadic CJD	Sporadic CJD	Sporadic CJD	Sporadic CJD	GSS (P102L)	Genetic CJD (E200K)
sex	male	male	female	male	male	male
codon 129	MV	MM	MM	MM	MM	MM
Typing of PrP-res	type 2	type 1	type 1	type 1	-	type 1
Age at onset (years)	69	70	59	62	43	66
Period from onset to death (months)	27	18	78	60	39	6
Period from onset to akinetic mutism (months)	3	2	3	4	25	3

CJD, Creutzfeldt-Jakob disease; GSS, Gerstmann-Sträussler-Scheinker syndrome; PrP-res, protease-resistant prion protein.

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
