# Peer review of "Postmortem Quantitative Analysis of Prion Seeding Activity in the Digestive System"

_molecules, 2019, doi:10.3390/molecules24244601_

Round 1
Reviewer 1 Report
The manuscript describes the detection of prion seeding activities from various tissues of the digestive system of human patients. The writing is clear and results are well presented.
I don't have major problems with the manuscript.
Some minor points:
the last sentence of the abstract: it is better to drop the word "infectious" (line 32) since there is no experiment really assaying the actual infectivity. I am also not sure about the word "similar" (line 33) since the seeding activity is still at least an order of magnitude smaller in esophagus samples compared with the brain, and only for 2 sCJD patients. Page 7: the style of references is not consistent.Author Response
Reviewer1
Some minor issues:
The last sentence of the abstract: it is better to drop the word "infectious" (line 32) since there is no experiment really assaying the actual infectivity. I am also not sure about the word "similar" (line 33) since the seeding activity is still at least an order of magnitude smaller in esophagus samples compared with the brain, and only for 2 sCJD patients.
In accordance with the reviewer’s comment, we have changed the expression in the indicated sentence.
Page 7: the style of references is not consistent.
We have changed the style of references to ensure consistency.
Please address all correspondence to: Katsuya Satoh, 1-7-1 Sakamoto, Nagasaki, 852-8501, Japan. Tel: +81-958-19-7991; E-mail: [email protected]
We appreciate your kind consideration of our manuscript, and look forward to hearing from you in due course.
Yours sincerely,
Katsuya Satoh, MD, PhD
Reviewer 2 Report
In the draft of the research article titled, “Postmortem quantitative analysis of prion seeding activity in the digestive system,” Katsuya Satoh and colleagues explore the prion seeding activity in tissue derived from various digestive organs in patients diagnosed with human prion diseases after autopsy. To evaluate the prion seeding activity, the researches rely heavily on the well-accepted and highly sensitive real-time quaking-induced conversion (RT-QuIC) assay. They conclude that prion seeding activity in digestive organ tissue can surpass that which they have previously determined to be noninfectious. This, as the researchers point out, is important because patients with human prion diseases often require endoscopic gastrostomy, which could lead to contamination of medical instruments and potentially spread of disease.
The methods in the article are straightforward and follow up on previous examination of nervous and peripheral organ tissue from the patient set diagnosed with sporadic CJD. Also, the assays used are well-accepted and their methods correspond with those used by others in this field. The experiments appear sufficiently rigorous.
One criticism involves the figures, which present the positive replicate % and concluded SD50 results, but do not present any of the RT-QuIC/ThT data from which these results have been derived.
The authors state, “Recent studies have shown that seeding activity in vitro, 54 determined by endpoint RT-QuIC, parallels the infectivity of prion-containing animal specimens.” Since the authors rely completely on this in vitro method, they should dedicate some additional text to RT-QuiC’s accuracy (or potential shortcomings) in identifying infectious PrPsc.
Some minor issues:
Did the authors define SD50 as the Log SD50/g of sample in their figures or in their results? I don’t remember seeing it.
Line 76 in section 2.1. Patients states “four female patients” with sCJD, but Table 1 only has one female. Which is correct?
Based on the results, the group proposes a potential for diagnosis via biopsy from upper digestive tissue while maintaining that the less invasive current test involving nasal brushings is accurate. Why would someone opt for a more invasive diagnostic method?
The group states that IHC revealed no abnormal PrP in any of the gut tissue, but provide no explanation. Further, IHC is not discussed in the methods and is not is not presented in any figures.
Line 47: “…. [PrPSc] in the spleen of this patient was approximately 10−4 higher than that in the brain tissue5”. In the article cited (reference 5), the authors observe PrPsc levels were approximately 10-4 less concentrated in spleens than in brains of sCJD patients.
Line 180: “Finally, Japanese doctors perform gastrostomy or tube feeding for some patients with CJD. These clinicians carefully treat the gastrostomy at the time of its creation”. These final sentences need elaboration. It’s not clear what the authors are saying specifically.
Author Response
Point-by-Point Responses to the Reviewers’ Comments
Reviewer 2
Some minor issues:
1.Did the authors define SD50 as the Log SD50/g of sample in their figures or in their results? I don’t remember seeing it.
We did not define SD50 as log SD50/g of sample in our figures or results. In accordance with the reviewer’s comment, we have added the following sentence: “all prion seeding activities for digestive organs (SD50) were defined as log SD50/g of tissue.”
2,Line 76 in section 2.1. Patients states “four female patients” with sCJD, but Table 1 only has one female. Which is correct?
We apologize for this error. We have corrected the number in the relevant sentence in the text (line 76).
3.Based on the results, the group proposes a potential for diagnosis via biopsy from upper digestive tissue while maintaining that the less invasive current test involving nasal brushings is accurate. Why would someone opt for a more invasive diagnostic method?
We tried the RT-QuIC assay with brushings from the nasal mucosa, but it was extremely difficult to detect. Other researchers have also tried to detect it unsuccessfully.
We also tried the QuIC assay with brushings from the nasal mucosa, but again it was extremely difficult to detect.
Meanwhile, GIS is a common test that includes a medical examination, and a gastric biopsy is one of the most common tests for gastroenterologists.
4.The group states that IHC revealed no abnormal PrP in any of the gut tissue, but provide no explanation. Further, IHC is not discussed in the methods and is not is not presented in any figures.
When formic acid treatment, as one of the pathological treatment processes for prion disease, is performed, the digestive organ tissue becomes extremely tattered to a level that it cannot be shown. For this reason, we did not present a figure.
5.Line 47: “…. [PrPSc] in the spleen of this patient was approximately 10−4 higher than that in the brain tissue5”. In the article cited (reference 5), the authors observe PrPsc levels were approximately 10-4 less concentrated in spleens than in brains of sCJD patients.
We have changed the expression in the indicated sentence.
6.Line 180: “Finally, Japanese doctors perform gastrostomy or tube feeding for some patients with CJD. These clinicians carefully treat the gastrostomy at the time of its creation”. These final sentences need elaboration. It’s not clear what the authors are saying specifically.
We have changed the indicated sentences as follows: Finally, Japanese doctors perform gastrostomy for some patients with CJD. When clinicians create the gastrostomy, they can pass deep into the stomach and abdominal walls. Therefore, clinicians treat the gastrostomy carefully at the time of its creation.